# Effects of CurraNZ, a New Zealand Blackcurrant Extract during 1 Hour of Treadmill Running in Female and Male Marathon des Sables Athletes in Hot Conditions: Two Case Studies

**DOI:** 10.3390/jfmk9020076

**Published:** 2024-04-18

**Authors:** Mark E. T. Willems, Patrick W. Bray, Holly M. Bassett, Tilly J. Spurr, Andrew T. West

**Affiliations:** Institute of Applied Sciences, University of Chichester, College Lane, Chichester PO19 6PE, UK; patrickwbray24@gmail.com (P.W.B.); hollythebassett@gmail.com (H.M.B.); m.spurr@chi.ac.uk (T.J.S.); a.west@chi.ac.uk (A.T.W.)

**Keywords:** Marathon des Sables, New Zealand blackcurrant, anthocyanins, exertional heat stress, treadmill running, substrate oxidation

## Abstract

Four weeks before competition in the 2023 Marathon des Sables, a 6-stage, ~250 km running event in the Sahara Desert, we examined the effects of a 7-day intake of New Zealand blackcurrant extract (210 mg anthocyanins per day) on 1 h treadmill running-induced physiological and metabolic responses in the heat (~34 °C, relative humidity: ~30%) in non-acclimatized amateur female and male athletes (age: 23, 38 yrs, BMI: 24.2, 28.4 kg·m^−2^, body fat%: 29.2, 18.8%, $\dot{V}$O_2max_: 50.1, 52.1 mL·kg^−1^·min^−1^). During the 1 h run at 50%$\dot{V}$O_2max_ (speed female: 7.3, male: 7.5 km·h^−1^), indirect calorimetry was used, and heart rate was recorded at 15 min intervals with core temperature monitoring (0.05 Hz). The 1 h runs took place 3 h after a light breakfast and 2 h after intake of the final dose of New Zealand blackcurrant extract with water allowed ad libitum during the run. The New Zealand blackcurrant extract had no effects on the female athlete. The respiratory exchange ratio (RER) of the female athlete in the non-supplement control condition was 0.77 ± 0.01, indicating an existing ~77% contribution of fat oxidation to the energy requirements. In the male athlete, during 1 h of running, fat oxidation was higher by 21% (*p* < 0.01), carbohydrate oxidation was 31% lower (*p* = 0.05), RER was 0.03 units lower (*p* = 0.04), and core temperature was 0.4 °C lower (*p* < 0.01) with no differences for heart rate, minute ventilation, oxygen uptake, and carbon dioxide production for the New Zealand blackcurrant condition compared to the non-supplement control condition. Seven-day intake of New Zealand blackcurrant extract (210 mg anthocyanins per day) provided beneficial physiological and metabolic responses during exertional heat stress by 1 h of indoor (~34 °C) treadmill running in a male Marathon des Sables athlete 4 weeks before competition. Future work is required to address whether New Zealand blackcurrant provides a nutritional ergogenic effect for Marathon des Sables athletes during long-duration running in the heat combined with personalized nutrition.

## 1. Introduction

The Marathon des Sables (six stages, ~250 km timed running race) is an ultra-endurance event in the Sahara Desert that requires physiological and metabolic adaptations (e.g., mitochondrial biogenesis, enhanced maximum oxygen uptake, and augmentation of lactate kinetics) by concurrent physical training and nutritional strategies. In addition, Marathon des Sables athletes have personality (e.g., extraversion and openness to experience) and psychological (e.g., mental toughness and confidence) attributes [1] that would allow them to cope with the required time-demanding preparatory training program and event competition. In general, in the last few weeks before an endurance or ultra-endurance running race, athletes will implement a tapering strategy that normally involves a reduction in the training volume with the intent to allow complete recovery of physical fatigue (for a systematic review and meta-analysis, see [2]). For the Marathon des Sables athletes who do not have the opportunity to train with exertional heat stress, a laboratory-based heat acclimatization program is commonly used to prepare for the hot-desert event [3,4] a few weeks beforehand to lower perceptions of heat stress and reduce thermoregulatory strain. Nevertheless, the physical training program adopted in preparation for the Marathon des Sables before a potential heat acclimation program will have provided the individual athlete with the essential physiological and metabolic adaptations required for competition in an ultra-endurance event.

In competition and exercise training, dietary supplement use is widespread and common among elite athletes (e.g., Japanese Olympic and Paralympic athletes [5], Spanish elite male and female athletes [6], gym members [7], recreationally active students [8], and ultramarathoners [9]). In addition, many position statements have provided evidence for the ergogenic nutritional potential of dietary supplements and nutritional considerations in different cohorts (e.g., [10,11]), including for single-stage ultra-endurance events [12]. Within the sports nutrition literature, only a few studies have examined the effects of supplementation in athletes during the Marathon des Sables. For example, Machefer et al. [13] observed that a daily multivitamin (containing e.g., 150 mg vitamin C and 8 mg vitamin E) and mineral supplementation by athletes (16 men and one woman) three weeks before and during the Marathon des Sables prevented lipid peroxidation quantified by plasma thiobarbituric acid reactive substance levels. This suggests that the training status of the ultra-endurance athletes and the competitive demands of Marathon des Sables do not preclude the effectiveness of antioxidant supplementation. In contrast, however, an intake over 12 weeks of a probiotic and glutamine powder was not effective in altering the extracellular heat shock protein 72 response [14].

In field and laboratory-based supplementation studies, it is common that the cohort or some individuals of the cohort can be considered non-responders to an ergogenic nutritional aid intervention (e.g., caffeine during a Wingate cycling test [15] and sodium citrate during 200 m swimming [16]). Training status may be one of the factors (for a systematic review on dietary nitrates, see [17], and for β-alanine, see [18]), but in an overall non-responding cohort to a nutritional intervention study, it is not common to reveal observations on responders. For example, recently, we observed with the acute intake of New Zealand blackcurrant extract that the ergogenicity during a 16.1 km time-trial was present in the slower cyclists but not in the faster cyclists [19]. Many studies have provided evidence for the effects of New Zealand blackcurrant extract in different cohorts and with different exercise modalities, maybe due to the antioxidant effects of the anthocyanin-induced plasma metabolites [20].

In sports nutrition studies, athletic ultra-endurance cohorts have never been laboratory-tested in the last few weeks before a competitive event. Such studies would be logistically challenging, and therefore, case studies can be warranted. For example, in an ultra-endurance male runner, a 7-day intake of anthocyanin-rich New Zealand blackcurrant extract in between 160 km running events altered the oxidation of carbohydrates and fat during 2 h of treadmill running (speed 10.5 km·h^−1^ covering a half-marathon distance) [21]. The observations by Willems and Briggs [21] were the first evidence that ultra-endurance athletes can respond to an anthocyanin-rich supplement. Interestingly, the enhanced exercise-induced fat oxidation in the ultra-endurance male athlete was similar to that of male cyclists with lower training status [22]. Marathon des Sables athletes compete in a hot-desert environment. No studies have examined the effectiveness of anthocyanin-rich New Zealand blackcurrant extract during controlled exertional heat stress in athletes in the weeks preceding the multi-stage Marathon des Sables ultra-endurance event. In addition, the physiological and metabolic effects of New Zealand blackcurrant extract have not been examined in female ultra-endurance athletes.

Therefore, the aim of the present study was to examine the effects of a 7-day intake of anthocyanin-rich New Zealand blackcurrant extract on the physiological and metabolic responses and core temperature in a female and a male amateur Marathon des Sables athlete during 1 h of treadmill running in hot conditions.

## 2. Materials and Methods

A female (age: 23 yr, BMI: 24.2 kg·m^−2^, body fat%: 29.2%, $\dot{V}$O_2max_: 50.1 mL·kg^−1^·min^−1^) and a male (age: 38 yr, BMI: 28.4 kg·m^−2^, body fat%: 18.8%, $\dot{V}$O_2max_: 52.1 mL·kg^−1^·min^−1^) amateur ultra-endurance athlete volunteered for the study. Participants were in preparation for the Marathon des Sables 2023 (23 April 2023–28 April 2023) and were invited for the study as both had contacted the University of Chichester (UK) for heat acclimation training. For both participants, it was their first time preparing for the Marathon des Sables. A session of heat acclimation exercise would normally last for about 60 to 90 min. In agreement with the participants, and in consideration of the aim of the study to examine the effectiveness of a 7-day intake of New Zealand blackcurrant extract, it was decided that before undertaking the 7-day heat acclimation, their physiological and metabolic responses would be measured during 1 h of treadmill running in the heat (~34 °C, relative humidity: ~30%) and compared to a non-supplement control condition. Ethical approval for the study was provided by the University of Chichester Research Ethics Committee (approval code: 2223_44, date of approval: 8 March 2023). Participants provided written informed consent following detailed information on the experimental procedures, measurements, potential risks, and the right to withdraw. Participants were considered healthy based on completion of a University-approved health history questionnaire and were not taking supplements that could be considered a confounding factor for the measurement of the physiological and metabolic responses during treadmill running. Participants were not heat-acclimatized and were tested for the effects of 7 days of intake of anthocyanin-rich New Zealand blackcurrant extract 3 to 4 weeks before participation in the Marathon des Sables 2023.

### 2.1. Experimental Design

The participants attended the exercise physiology laboratory at the University of Chichester for three visits (details described below). In short, in the first visit (15 March 2023) at ambient temperature (~19 °C), anthropometric, physiological, and body composition parameters were recorded. Subsequently, the participants performed an incremental submaximal treadmill running protocol and an incremental treadmill running protocol to exhaustion. Protocols were used to determine maximum oxygen uptake ($\dot{V}$O_2max_) and the running speed at 50%$\dot{V}$O_2max_. In visits two (control, 24 March 2023) and three (New Zealand blackcurrant (NZBC) extract, 31 March 2023), the participants performed the 1 h treadmill run, and core temperature (0.05 Hz) and physiological and metabolic parameters were measured every 15 min. Visits two and three were morning visits and 30 days and 23 days before the start of the Marathon des Sables 2023. The participants abstained from strenuous exercise and alcohol for 48 h prior to each laboratory visit but were allowed to continue with their habitual training program but advised to limit exercise the day before testing. Dietary intake was recorded for 48 h prior to visits two (control) and three (NZBC extract) (table in Section 3.1), and habitual anthocyanin intake was quantified with a food frequency questionnaire with foods and drinks listed in the Phenol-Explorer database [23].

### 2.2. Visit One—Preliminary Measurements

Following the measurements of stature (Holtain Stadiometer, Crymych, Dyfed, UK) and body mass (Seca Model 876, Seca Ltd., Birmingham, UK), the participants were seated for 10 min. Blood samples were taken with the finger prick method for resting hematocrit (Hawksley MicroHaematocrit Reader, Hawksley and Son Ltd., Lancing, West Sussex, UK), hemoglobin (HemoCue 201+, HemoCue, Dronfield, Derbyshire, UK), glucose, and lactate (Biosen C-line, EKF diagnostics, Cardiff, UK). The participants then completed an incremental submaximal test and an incremental test to volitional exhaustion (for details see below).

#### Incremental Submaximal Test and Incremental Test to Exhaustion

The incremental submaximal test consisted of 8 stages of 4 min on a motorized treadmill (Woodway ELG70, Cranlea & Co, Birmingham, UK). The treadmill gradient was set at a 1% incline [24]. The starting treadmill speed was 8 km∙h^−1^ with subsequent increments of 0.75 km∙h^−1^ for each stage. In the final 90 s of each stage, the expired air was collected as the participants were wearing a mouthpiece connected with tubing to Douglas bags. The linear relationship between treadmill running speed and the oxygen uptake responses during the incremental submaximal test allowed the quantification of the running speed that elicits 50%$\dot{V}$O_2max_. A treadmill running speed at 50%$\dot{V}$O_2max_ was used for the control and New Zealand blackcurrant extract visits. After the incremental submaximal test, the participants completed an incremental treadmill running protocol to volitional exhaustion for the measurement of maximum oxygen uptake. For this test, the treadmill gradient was set at 1% with a starting speed of 8 km∙h^−1^ and increments of 0.1 km∙h^−1^ every 6 s until volitional exhaustion. Expired air was collected in the final 4 min of the test with Douglas bags. At volitional exhaustion, the treadmill speed and heart rate for the female and male athletes were 16.56 and 18.54 km∙h^−1^ and 193 and 188 beats∙min^−1^ (recorded with short-range telemetry, RS400, Polar Electro UK Ltd., Warwick, UK). For the measurements of respiratory parameters, the expired air was analyzed for fractions of oxygen and carbon dioxide using a calibrated paramagnetic oxygen analyzer and an infrared carbon dioxide analyzer (series 1440; Servomex plc, Crowborough, UK). The volume of expired air was measured with a calibrated dry gas meter (Harvard Apparatus Ltd., Edenbridge, UK) with simultaneous measurement of expired air temperature during Douglas bag evacuation for gas volume temperature corrections. Gas volumes were expressed as values with standard temperature, standard barometric pressure, and dry.

### 2.3. Supplementation Strategy

The visits for the control and NZBC extract condition did have the recording of physiological, metabolic, and thermoregulatory responses. As far as we know, there is no bias possible during the measurements. In addition, the aim was to have the participants tested as close to the event as possible and before undertaking the one-week heat acclimatization program. Seven days before the visit with New Zealand blackcurrant extract (i.e., 23 days before the start of the Marathon des Sables), the participants ingested 600 mg of New Zealand blackcurrant extract (CurraNZ™, Health Currancy Ltd., Surrey, UK) every day. The athletes consumed two capsules per day at breakfast, with each capsule containing 105 mg of anthocyanins. The anthocyanin composition of the capsules was provided by the company and consisted of 35–50% delphinidin-3-rutinoside, 5–20% delphinidin-3-glucoside, 30–45% cyanidin-3-rutinoside, and 3–10% cyanidin-3-glucoside 3–10%.

### 2.4. Control and New Zealand Blackcurrant Conditions Visits

Upon arrival for the visits with the 1 h treadmill run, hydration status was determined with a measurement for urine osmolality (Osmocheck PAL-OSMO; Vitech Scientific, Partridge Green, West Sussex, UK). Body mass was recorded before and after the 1 h run. Participants inserted a polyethylene rectal thermistor (Edale Instruments, Cambridge, UK) for the measurement of core body temperature (0.05 Hz) and entered the environmental chamber (TISS Model 201003-1, TIS Services UK, Medstead, Hampshire, UK). Environmental chamber conditions for temperature and humidity were ~34 °C and 30%. After 10 min of rest, the treadmill run (1% gradient [24]) was initiated. The treadmill speeds for the 1 h run at 50%$\dot{V}$O_2max_ were 7.3 km·h^−1^ for the female and 7.5 km·h^−1^ for the male athlete. Heart rate measurements and expired air collections with Douglas bags and fractions of oxygen and carbon dioxide in the environmental chamber were taken every 15 min during the 1 h run. During the 1 h run in visits two and three, the participants were allowed to drink water ad libitum at self-selected times.

### 2.5. Data Calculations and Statistical Analysis

Substrate oxidation during the 1 h treadmill run was calculated with the proposed equations from Jeukendrup and Wallis [25] for exercise with moderate intensity (50%$\dot{V}$O_2max_) to high intensity (75%$\dot{V}$O_2max_). For analysis of the physiological (i.e., heart rate, minute ventilation, oxygen uptake, carbon dioxide production) and metabolic parameters (i.e., fat oxidation, carbohydrate oxidation, and respiratory exchange ratio) in the control and New Zealand blackcurrant extract conditions, a paired two-tailed *t*-test was used (GraphPad Prism v5 for Windows, GraphPad software, San Diego, CA, USA) for the 15 min (i.e., 4 time points) during the 1 h treadmill run. Core temperature recordings were averaged for 15 min time periods and analyzed with a paired two-tailed *t*-test. Data are reported as mean ± SD, calculated from the 15 min (i.e., 4 time points) measurements during the 1 h treadmill run. Significance was accepted at *p* < 0.05.

## 3. Results

Table 1 provides the participants’ characteristics and physiological and body composition parameters of the female and male amateur Marathon des Sables athletes.

### 3.1. Dietary and Energy Intake and Habitual Anthocyanin Intake of Female and Male Marathon des Sables Athletes

The female and male amateur Marathon des Sables athletes did not receive support for dietary intake during training in preparation for the event. Table 2 reports the dietary and energy intake of the female and male Marathon des Sables athletes. Both athletes had relatively high fat and protein intake. Habitual dietary anthocyanin intake (i.e., without the intake of New Zealand blackcurrant extract) for the female and male Marathon des Sables athletes was estimated to be 47.3 and 91.5 mg·day^−1^, respectively.

### 3.2. Female Marathon des Sables Athlete

#### 3.2.1. Urine Osmolality, Water Intake, and Body Mass Changes

On arrival for the 1 h run, urine osmolality was 720 mOsm·kg^−1^ in the control condition and 350 mOsm·kg^−1^ in the New Zealand blackcurrant extract condition. Ad libitum water consumption during the 1 h treadmill run was 125 mL (control condition) and 175 mL (New Zealand blackcurrant extract condition). Body mass change in the 1 h treadmill run was −1.16 kg (i.e., −1.76%) in the control condition and −0.53 kg (i.e., −0.80%) in the New Zealand blackcurrant condition.

#### 3.2.2. Physiological Responses

During the 1 h treadmill run, there were no effects of anthocyanin-rich New Zealand blackcurrant extract on heart rate (control: 151 ± 12, NZBC extract: 152 ± 14 beats·min^−1^, *p* = 0.24), minute ventilation (control: 42.6 ± 1.9, NZBC extract: 44.0 ± 2.4 L·min^−1^, *p* = 0.33), oxygen consumption (control: 1.80 ± 0.06, NZBC extract: 1.75 ± 0.04 L·min^−1^, *p* = 0.19), carbon dioxide production (control: 1.39 ± 0.06, NZBC extract: 1.35 ± 0.02 L·min^−1^, *p* = 0.28), or core temperature (control: 38.2 ± 0.6, NZBC extract: 38.1 ± 0.6 °C, *p* = 0.10) for the female Marathon des Sables athlete.

#### 3.2.3. Metabolic Responses

During the 1 h treadmill run, there were no effects of anthocyanin-rich New Zealand blackcurrant extract on fat oxidation (control: 0.68 ± 0.03, NZBC extract: 0.67± 0.04 g·min^−1^, *p* = 0.30), carbohydrate oxidation (control: 0.56 ± 0.09, NZBC extract: 0.52 ± 0.05 g·min^−1^, *p* = 0.54), and respiratory exchange ratio (control: 0.77 ± 0.01, NZBC extract: 0.77 ± 0.01, *p* = 0.72) for the female Marathon des Sables athlete. The female Marathon des Sables athlete may not have responded to the intake of New Zealand blackcurrant extract as the female athlete already had a low respiratory exchange ratio in the control condition, indicating a high intrinsic ability, perhaps from the physical training, to use fat as an energy source.

### 3.3. Male Marathon des Sables Athlete

#### 3.3.1. Urine Osmolality, Water Intake, and Body Mass Changes

On arrival for the 1 h run, urine osmolality was 770 mOsm·kg^−1^ in the control condition and 690 mOsm·kg^−1^ in the New Zealand blackcurrant extract condition. Ad libitum water consumption during the 1 h treadmill run was 105 mL in the control condition and 210 mL in the New Zealand blackcurrant extract condition. Body mass change by the 1 h treadmill run was −1.55 kg (i.e., −1.65%) in the control and −1.19 kg (i.e., −1.26%) in the New Zealand blackcurrant condition.

#### 3.3.2. Physiological Responses

During the 1 h treadmill run, there were no effects of anthocyanin-rich New Zealand blackcurrant extract on heart rate (control: 136 ± 10, NZBC extract: 134 ± 7 beats·min^−1^, *p* = 0.50), minute ventilation (control: 51.2 ± 1.8, NZBC extract: 49.8 ± 3.1 L·min^−1^, *p* = 0.44), oxygen consumption (control: 2.51 ± 0.02, NZBC extract: 2.60 ± 0.09 L·min^−1^, *p* = 0.10), or carbon dioxide production (control: 2.01 ± 0.05, NZBC extract: 1.99 ± 0.04 L·min^−1^, *p* = 0.71). Core temperature was 0.4 °C lower (control: 37.7 ± 0.3, NZBC extract: 37.3 ± 0.3 °C, *p* < 0.01).

#### 3.3.3. Metabolic Responses

During the 1 h treadmill run, New Zealand blackcurrant extract enhanced fat oxidation by 21% (*p* = 0.009) (Figure 1a). Carbohydrate oxidation was 31% lower (*p* = 0.05) (Figure 1b). The respiratory exchange ratio was lower by 0.03 units (control: 0.80 ± 0.02, NZBC extract: 0.77 ± 0.01; *p* = 0.04).

## 4. Discussion

The main finding of the present study was that the male Marathon des Sables athlete responded to the intake of New Zealand blackcurrant extract with similar metabolic responses, i.e., an increase in exercise-induced fat oxidation and a decrease in exercise-induced carbohydrate oxidation, as was also observed in a male endurance-trained cohort during cycling (intake 7 days of 105 mg anthocyanins·day^−1^) [22] and recreationally active males during treadmill running in the heat (intake 7 days of 210 mg anthocyanins·day^−1^) [26]. Case studies on the effects of nutraceutical supplementation on physiological and metabolic responses in physically trained ultra-endurance athletes before event competition are uncommon. In Willems and Briggs [21], an amateur male ultra-endurance athlete responded in the 5 weeks between 100-mile running events to anthocyanin-rich New Zealand blackcurrant extract with enhanced fat oxidation during 2 h of treadmill running (speed: 10 km·h^−1^, 26 °C). In the present study, the male Marathon des Sables athlete was going to compete for the first time in the event and was undergoing the required physical training without personal training support. Therefore, it is possible that the physical training strategy did not provide the optimal adaptation of key enzymes and structural proteins involved in lipolysis, membrane transport of free fatty acids, and intracellular mitochondrial processes [27]. This less-than-optimal adaptation would allow anthocyanin-induced metabolites to affect one or more of the many events involved in exercise-induced fat oxidation. However, case studies with a focus on physiological and metabolic responses to the intake of New Zealand blackcurrant extract or any other dietary supplementation can easily provide a non-response, as is common in cohort studies that examine the effects of supplementation (e.g., Margaritelis et al. [28]). Therefore, the effectiveness of the intake in the present study for the male Marathon des Sables athlete during exertional heat stress will require confirmation in a cohort study of male athletes and with an adequate sample size [29] to confirm generalizability. Similar to the observation in Cook et al. [22] and Hiles et al. [26], the male athlete did not respond with physiological changes, e.g., no change in heart rate and oxygen uptake. However, the male athlete had a thermoregulatory response with a temperature that was lower by 0.4 °C during the treadmill run. The latter response may be due to an alteration of blood flow to the skin due to the vasodilatory effects of anthocyanins and anthocyanin-induced metabolites [30,31].

In contrast to the responsiveness of the male athlete, the present study showed that the female Marathon des Sables athlete did not respond to the intake of anthocyanin-rich New Zealand blackcurrant extract. This was in contrast with cohort observations in female endurance-trained cyclists (7 days’ intake of 210 mg anthocyanins·day^−1^) [32] and recreationally active females (7 days’ intake of 210 mg anthocyanins·day^−1^) [33]. It is possible that the absence of a blackcurrant-induced effect on metabolic responses in the female Marathon des Sables athlete was due to the athlete having a low respiratory exchange ratio of 0.77. Women tend to have lower respiratory exchange ratio values during moderate-intensity aerobic exercise [34] and low-level isometric exercise [35]. In addition, training status is a potent modulator of the respiratory exchange ratio [36].

With respect to ultra-endurance activities, the sex differences in physiology may benefit females (for a review, see Tiller et al. [37]). However, in males and females that were performance-matched for 42.2 km running, the better 90 km performance in females was not associated with enhanced fat metabolism [38]. Interestingly, for an Ironman (i.e., 3.9 km (2.4 mile) swimming, 180.2 km (112 mile) cycling, and 42.2 km (26.2 mile) running), peak fat oxidation in females [39] and males [40] was associated with race time. The importance of exercise-induced fat oxidation for the race time of a Marathon des Sables athlete is not known. Future studies in cohorts of females and males should examine whether the intrinsic ability to have high fat oxidation—intrinsic or from physical training—is a determinant for the effectiveness of anthocyanin-rich New Zealand blackcurrant to affect metabolic responses. In addition, it is possible that individuals with a very low respiratory exchange ratio need a longer or higher intake of anthocyanin-rich New Zealand blackcurrant extract to affect metabolic responses. In addition, future work is needed to address the effect of anthocyanin-rich New Zealand blackcurrant in Marathon des Sables athletes during exercise conditions in which fueling strategies adopted by the athletes are incorporated.

Several limitations of the present study need to be noted. First, the reliability of the physiological, metabolic, and core temperature responses in our participants was not known. In general, for the measurement of reliability, factors that could potentially contribute to the variation of the measurements should be controlled. We could not interfere with their scheduled individual training activities in the weeks before experimental testing to allow reliable measurements of the physiological, metabolic, and core temperature responses. Second, although our participants were advised to limit exercise on the day before experimental testing, we cannot exclude that participants were tested in a fatigued state.

## 5. Conclusions

From the present observations, it is concluded that the male novice Marathon des Sables athlete responded with metabolic changes (e.g., enhanced exercise-induced fat oxidation) to the 7-day intake of anthocyanin-rich New Zealand blackcurrant extract during a 1 h treadmill run in the heat. Such changes were not observed in the female novice Marathon des Sables athlete, maybe due to already having high intrinsic exercise-induced fat oxidation. Cohort studies in Marathon des Sables athletes are needed to confirm the generalizability of our findings.

## Figures and Tables

**Figure 1 jfmk-09-00076-f001:**
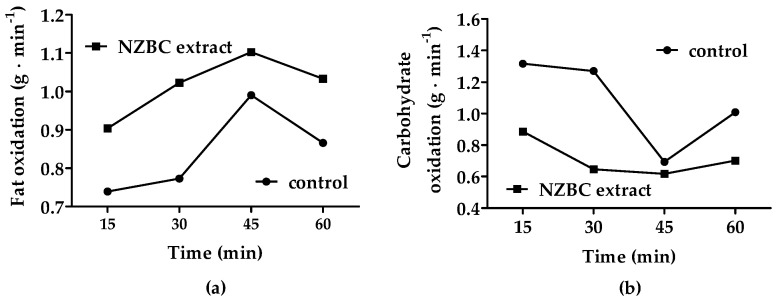
Substrate oxidation of the male Marathon des Sables athlete during 1 h of treadmill running (speed: 7.5 km·h^−1^) in the heat (~34 °C, relative humidity: ~30%): (**a**) fat oxidation; (**b**) carbohydrate oxidation.

**Table 1 jfmk-09-00076-t001:** Participants, body composition, and physiological characteristics of the female and male Marathon des Sables athletes.

Parameters	Female	Male
Age (years)	23	38
Height (cm)	165	182
Body mass (kg)	65.5	94.2
BMI (kg·m^−2^)	24.2	28.4
Skeletal muscle mass (kg)	25.8	43.7
Body fat%	29.2	18.8
Body fat mass (kg)	19.1	17.7
Hematocrit rest (%)	46%	48%
Hemoglobin rest (g·dL^−1^)	15.9	16.3
Blood lactate rest (mMol·L^−1^)	1.12	1.71
Glucose rest (mMol·L^−1^)	4.50	4.51
$\dot{V}$O_2max_ (mL·kg^−1^·min^−1^)	50.1	52.1

BMI, body mass index.

**Table 2 jfmk-09-00076-t002:** Dietary and energy intake 24 h before each experimental visit for the female and male Marathon des Sables athletes.

Female Marathon des Sables Athlete		
	Conditions	
	control	NZBC extract
parameters		
Carbohydrate (g, %)	157, 49	138, 34
Fat (g, %)	165, 30	182, 41
Protein (g, %)	68, 21	98, 25
Energy intake (kcal)	1294	1592
Male Marathon des Sables Athlete		
	Conditions	
	control	NZBC extract
parameters		
Carbohydrate (g, %)	161, 28	96, 20
Fat (g, %)	100, 39	99, 40
Protein (g, %)	187, 33	150, 31
Energy intake (kcal)	2297	1987

NZBC, New Zealand blackcurrant.

## Data Availability

Data are available on reasonable request.

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
