# Peer review of "Effects of CurraNZ, a New Zealand Blackcurrant Extract during 1 Hour of Treadmill Running in Female and Male Marathon des Sables Athletes in Hot Conditions: Two Case Studies"

_jfmk, 2024, doi:10.3390/jfmk9020076_

Round 1

Reviewer 1 Report

Comments and Suggestions for Authors

Willems_Curra_jfmk_2024. Reviewer report

Thank you for the opportunity to review this manuscript, it is on an interesting topic. 

I have some minor concerns highlighted bellow. 

General comments. 

Please, change the “P value” to “p value” in the whole text. 

Specific comments

Title. Please add: a two clinic cases study. 

Abstract

Line 19. Please indicate that the changes in the male athlete were with respect a control measurement. 

Introduction

Line 85. Please adds some sentence explaining the potential beneficial effects of anthocyanin in sports. 

Reviewer 2 Report

Comments and Suggestions for Authors

L 121: seems like Ex Phy would be lowercased.  Is it a proper name?

Table 2: should there be subheaders under parameters to note the different sessions?

Conclusion

I think this needs revision.  Your case report shows the potential; I would argue that it is not "concluded"; b/c we see in this paper that the female subject didn't have any response.  And you note earlier in the paper that sometimes some people don't have responses to supplement studies.  Tempering the language is most appropriate

Reviewer 3 Report

Comments and Suggestions for Authors

This study analyzed the effect of anthocyanin-rich New Zealand blackcurrant extract on physiological responses during high-temperature running. It can be used as valuable information in sports nutrition to improve marathon performance. However, supplementation and correction are required for the following matters.

In the introduction, please add detailed information on the components of anthocyanin-rich New Zealand blackcurrant extract and previous research results on its effect on marathon performance.

In the discussion, since there is no group that analyzed the effect of ingestion at room temperature, it is believed that there is a need to add description of the limitations of the study or the estimated results in this area.

Reviewer 4 Report

Comments and Suggestions for Authors

Dear authors,

This study was conducted to effects of 7-day intake of New Zealand blackcurrant extract (210 mg anthocyanins per day) on 1 h treadmill running-induced physiological and metabolic responses in the heat (~34°C, relative humidity: ~30%) in a non-acclimatized amateur female and male athlete.

Although this study has strength on exertional heat stress which a laboratory-based heat acclimatization program, the participants consist of only one man and one woman. As we know, it is impossible to acknowledge the scientific reliability and validity of this study, which was conducted with only one participant. And, the measurement variables (fat oxidation and carbohydrate oxidation) was very well known topic internationally and this topic can be seen as generalized in sports science fields. For this reason, the academic quality of these variables (fat oxidation and carbohydrate oxidation) used in this study is very low scientific sounds. Furthermore, this proves that the study is not original enough. I do not think that this study can add innovation to the literature.

Unfortunately, this paper presents methodological falls, such as: the purpose is vague, there is no data reliability, and a predictive analysis were applied, statistical analysis is missing a lot of information. For this reason, I must say that the manuscript lacks clarity and depth in several areas.

Comments on the Quality of English Language

Moderate editing of English language required

Round 2

Reviewer 4 Report

Comments and Suggestions for Authors

The data analysis quality in this report is insufficient for publication. It does not meet the scientific standards. This is an observational study which is susceptible to selection bias and confounding. The authors analyzed their study like a randomized controlled trial without controlling for important confounding factors. The observed associations are not valid.

Comments on the Quality of English Language

Moderate editing of English language required